# Characterization of *Alternaria* Species Associated with Black Spot of Strawberry in Dandong, China

**Xiaozhe Sun** [1,†]**, Cuiyan Wang** [1,†]**, Xu Gao** [1]**, Xuehong Wu** [2,*] **and Yu Fu** [3,*]

1  College of Chemistry and Life Science, Anshan Normal University, Anshan 114007, China
2  Department of Plant Pathology, China Agricultural University, Beijing 100193, China
3  Department of Chemistry, Yanbian University, Yanji 133002, China
*  Correspondence: wuxuehong@cau.edu.cn (X.W.); fuyu@ybu.edu.cn (Y.F.)
†  These authors contributed equally to this work.

**Abstract:** Dandong has become the largest strawberry production and export base in China. Strawberry black spot disease is widespread and causes significant economic losses to strawberry growers in both the growing and harvest seasons. Until now, no study has reported the presence of the *Alternaria* species, the pathogen of strawberry black spot disease, in Dandong, Liaoning province, China. In 2020–2022, 108 isolates were obtained from strawberry leaves with typical symptoms of strawberry black spot disease from 56 major professional growing operations. Combined with morphological and molecular characteristics, the majority of isolates were identified as *A. tenuissima* (78 isolates, 72.2%), which had established total supremacy, followed by *A. alternata* (30 isolates, 27.8%). The pathogenicity results show that *A. tenuissima* and *A. alternata* are the two main pathogenic factors of strawberry black spot disease, the disease indexes of which were designated as 49.6–100.0% and 20.4–59.5%. To our knowledge, this paper is the first to identify *A. tenuissima* and *A. alternata* as causing black spot disease in strawberries in Dandong, China.

**Keywords:** Fragaria × ananassa "Red Face"; black spot; histone 3 gene; *Alternaria alternata*

## 1. Introduction

The Dandong 99 strawberry is a kind of "Benihoppe" Strawberry (Fragaria × ananassa "Red Face"). Thanks to the inherent advantages of being located in an internationally recognized high-quality fruit zone at a 40° north latitude, Dandong 99 strawberries are perfect in color, flavor, and taste, and are loved by consumers locally and abroad [1]. In 2021, the area dedicated Dandong 99 strawberry production extended to more than 1.28 million hectares, the total output reached 300,000 tons, the output value exceeded USD 93.95 million, the annual export of strawberry products was nearly 40,000 tons, and the export of foreign exchange was USD 35 million [2]. Dandong has become the largest strawberry production and export base in China.

Dandong 99 strawberry cultivation began in 1924 and mainly takes place in solar greenhouses in Dandong. Because of the high humidity and continuous cropping that takes place here, the strawberries are especially susceptible to disease. When the leaves display properties of being 3–5 mm in diameter and irregular, with brown circular lesions and sometimes with a yellowish halo, the strawberries are thought to be infected with black spot disease [3]. Strawberry black spot disease is widespread and causes significant economic losses to strawberry growers, both in the growing season and in the harvest season.

*Alternaria*, the main pathogenic fungus species, has more than 4000 recognized members, including nearly 300 species found globally [4]. This makes it particularly difficult to identify *Alternaria* species. With the development of molecular technology, the combination of morphology and molecular biology is being used in the identification of *Alternaria* species [5]. *Alternaria alternata* has been identified as the pathogen causing strawberry black spot disease in New Zealand [6], Japan [7], Korea [8], Italy [9], and Pakistan [10]. Later,

*A. tenuissima* was also found in Iran [11], Korea [12], and Taiwan [13]. Previous studies have shown that *Alternaria alternata* and *A. tenuissima* are the main causative agents of strawberry black spot disease in Beijing, China [14]. Many environmental factors affect the growth of *Alternaria* species [15]. At present, the phylogenetic relations and phenotypic characteristics of the *Alternaria* species causing strawberry black spot disease are under-explored in Dandong.

Interpreting the composition of *Alternaria* species is conducive to the prevention and treatment of strawberry black spot disease in Dandong. In this study, *Alternaria*'s composition and disease-causing capability are assessed for Dandong, in the Liaoning province of China.

## 2. Materials and Methods

### 2.1. Fungal Isolation and Morphological Characteristics

From 2020 to 2022, diseased leaves of strawberries were collected throughout the city of Dandong, Liaoning province, China (Table 1). After the samples were collected, they were placed into a 4-degree sampling incubator and brought to the laboratory for strain separation. Each isolate was measured to ascertain its conidial characteristics, body and beak length, shape, and the number of longitudinal and transverse septa; after this, they were incubated on potato carrot agar (PCA) for 7 days [16]. The length of conidial chains and the branching type were recorded with a stereo microscope (Leica M165C). The phenotypes of isolates grown on potato dextrose agar (PDA) and PCA plates were compared with those of the reference strains of *A. alternata*, *A. tenuissima*, and *A. arborescens* [17].

**Table 1.** Geographic origins and numbers of Alternaria isolates derived from strawberry leaves with typical features of black spot disease in Dandong, Liaoning province, China.

| Geographic Districts of Dandong | Number of Fields | Number of *Alternaria* Isolates | |
| --- | --- | --- | --- |
| | | *A. tenuissima* | *A. alternata* |
| Zhenxing District, Tangshancheng Town | 1 | 4 (2) [a] | 2 (1) |
| Zhenxing District, Wulongbei Town | 1 | 3 (1) | 0 (0) |
| Donggang City, Yiquan Town | 20 | 15 (5) | 6 (2) |
| Donggang City, Longwangmiao Town | 3 | 4 (2) | 2 (1) |
| Donggang City, Helong Town | 4 | 6 (3) | 2 (1) |
| Donggang City, Majiadian Town | 9 | 8 (4) | 5 (2) |
| Donggang City, Changshan Town | 4 | 6 (2) | 3 (1) |
| Donggang City, Beijingzi Town | 3 | 4 (2) | 2 (1) |
| Donggang City, Xiaodianzi Town | 3 | 5 (3) | 4 (2) |
| Donggang City, Qianyang Town | 2 | 6 (2) | 3 (1) |
| Donggang City, Huangtukan Farm | 4 | 8 (3) | 1 (1) |
| Fengcheng City, Baoshan Town | 1 | 5 (2) | 0 (0) |
| Fengcheng City, Hongqi Town | 1 | 4 (2) | 0 (0) |
| Total | 56 | 78 (33) | 30 (13) |
| Ratio | – | 72.2% | 27.8% |

[a] Numbers in the parentheses represent the numbers of corresponding *Alternaria* isolates subjected to pathogenicity tests.

### 2.2. DNA Extraction and Phylogenetic Analysis

Genomic DNA was extracted from mycelium harvested from the colonies of 108 isolates grown on PDA for 7 days, following the CTAB method described previously [18]. Partial sequences of the internal transcribed spacer (ITS) rDNA region were firstly performed for molecular identification. A 25 μL reaction volume containing 0.5 μL of ITS1 (5′-TCTGTAGGTGAACCTGCGGG-3′), 0.5 μL of ITS4 (5′-TCCTCCGCTTA TTGATATGC-3′), 10.5 μL of dd $H_2O$, 12.5 μL of Premix Ex Taq (v. 2.0, TaKaRa—containing 0.625 U Taq DNA polymerase, 200 μM dNTP, and 2 mM $MgCl_2$), and 1 μL of genomic DNA was subjected to polymerase chain reaction (PCR) amplification, according to the method of Fu et al. [14]. The PCR products were sequenced by Beijing Tianyihuiyuan Biotech



Co., Ltd. The obtained sequences were edited with DNAMAN (Version 5) and we identified the published homology from the National Center for Biotechnology Information (NCBI) library. The results show that 108 isolates belonged to the *Alternaria* species *A. alternata* and *A. tenuissima.* To further differentiate the samples, the partial coding sequences of the histone 3 gene were placed in a 25 μL reaction volume containing 0.5 μL of H3-1a (5′-ACTAAGCAGACCGCCCGCAGG-3′) and 0.5 μL of H3-1b (5′-GCGGGCGAGCTGGATGTCCTT-3′). After validation, Clustal W (version 1.83) [19] and MEGA5 program version 5.2.2 (http://www.megasoftware.net/ Access on 11 December 2022) were applied to establish the homological relationship of 108 *Alternaria* isolates using the Maximum Likelihood method (ML).

*2.3. Pathogenicity Analysis*

In order to assess the pathogenicity of the isolates, 46 *Alternaria* isolates were selected in vitro using the conidia inoculation method with some modifications [20]. Briefly, 46 *Alternaria* isolates were incubated on PDA plates at 25 °C for 7 days. Then, the fresh healthy leaves of 45 growing strawberry plants of the variety "Dandong 99" were collected from Yiquan Town in the city of Donggang, China. Agar plugs (5 mm in diameter) from the edges of the colonies of 46 *Alternaria* isolates were inserted on one side of the leaves. In total, 30 of the leaves were placed into one group isolation. After incubation in a growth chamber at 25 °C and 90% RH with a 12 h per day fluorescent light photoperiod for 14 days, the status of the detached leaves was observed. Disease severity (DS) was scored on a 5-point rating system derived, with modifications, from Pryor and Michailides (2002) [20], whereby 0 = no lesion, 1 = lesion < 1 mm in diameter, 2 = lesion 1 to 5 mm in diameter, 3 = lesions 5 to 10 mm in diameter, and 4 = lesions > 10 mm in diameter. Disease index (DI) values were calculated according to the method of Fu et al. [14], whereby DI = [100 × $\sum$ (n × corresponding DS)]/(N × 4), where n is the number of infected inoculation sites corresponding to each disease rating, and N is the total number of inoculation sites. Statistical significance was identified at $p < 0.05$, using the least significant difference test. All the diseased tissues were re-isolated and assessed in relation to Koch's postulates.

**3. Results**

*3.1. Isolation and Identification of Alternaria Species Associated with Strawberry Black Spot*

In 2020–2022, 108 isolates were obtained from strawberry leaves with typical symptoms of strawberry black spot disease from 56 major strawberry growing operations (Table 1).

The colonies of 78 isolates, identified as *A. tenuissima*, were initially greyish green to olive brown when assessed on the PDA plates. The conidiophores were singular, short, and measured 14.9–62.5 by 3.1–6.5 μm in size. The spores were generally ovoid or obclavate, and 15.0–47.1 by 7.5–18.6 μm in size. Here, 12 spores, comprising one to four transverse and zero to two longitudinal septa, constituted a linear chain, each of which was deeply nested among one or two lateral branches on the PCA plates (Figure 1a,c,e). The colonies of 30 isolates identified as *A. alternata* changed from terra brown to an iron coloring on the PDA plates. There were multiple straight conidial chains, which produced 8–12 spores. The conidiophores were 18.1–53.8 by 3.0–7.1 μm in size. The spores were generally egg-shaped or obclavate, and they were deeply nested among one to five transverse and zero to two longitudinal septa. The spores were 18.7–44.3 by 7.0–16.8 μm in size (Figure 1b,d,f). All the conidial chains, spores, and growth characteristics described here are consistent with those of *A. tenuissima* and *A. alternata* as described by Simmons [17].

The PCR products of 108 *Alternaria* isolates were 570 bp, amplified by the universal primers ITS1 and ITS4, and showed over 100% homology to those of *A. tenuissima* (GenBank Accession Nos. KR867207) and *A. alternata* (GU566303, KR867035, and MH862229). After amplification by the histone 3 gene primer, the PCR products of 108 isolates were divided into two fragments: 546 bp (78 isolates) and 440 bp (30 isolates). The phylogenic results show that 78 *Alternaria* isolates fit into Group I, which was compatible with 99%

of *A. tenuissima* (AF404634, JX495167, and JX495168) based on the histone 3 gene (partial cds). All sequences of the 108 isolates are listed in Supplementary Table S1. Two rootless phylogeny trees were created for the 46 *Alternaria* isolates used in the pathogenicity experiments, based on ITS sequence and the histone 3 gene data (Figures 2 and 3).

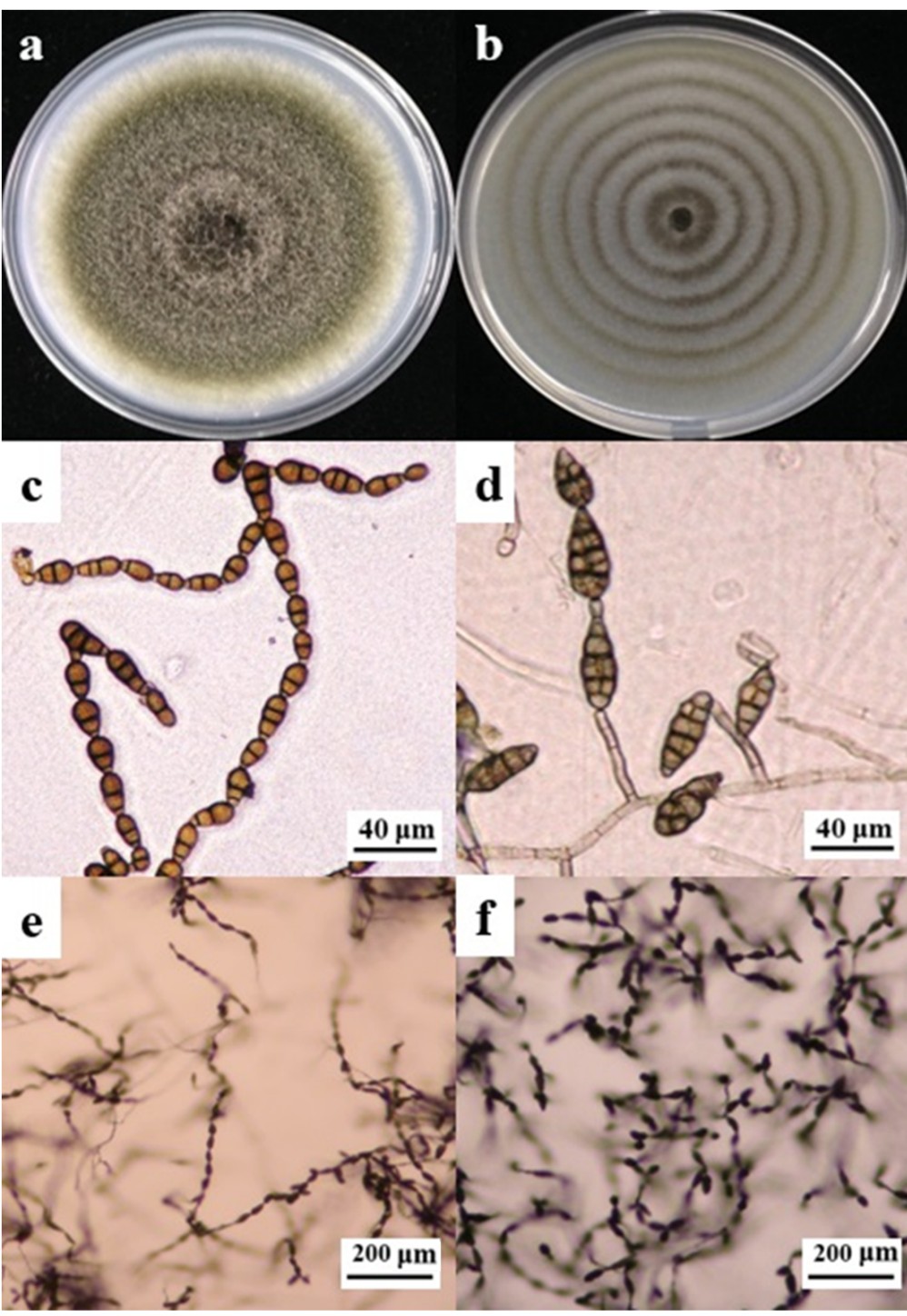

**Figure 1.** Morphology of the representative isolates of *Alternaria tenuissima* and *A. alternata*. (**a**): colonies of *A. tenuissima*; (**b**): colonies of *A. alternata*; (**c**): conidia of *A. tenuissima*; (**d**): conidia of *A. alternata*; (**e**): sporulation patterns of *A. tenuissima*; (**f**): sporulation patterns of *A. alternata*.

**Figure 2.** Phylogenetic tree based on the partial coding sequences of the ITS gene of the 46 *Alternaria* isolates used in the pathogenicity tests and the 8 reference sequences published in the NCBI library. The algorithm is based on the Maximum Likelihood estimate. Bootstrap values with 1000 replications were found at nodes on the phylogenetic tree, which identified the *A. brassicae* isolate AB11, *A. cucumerina* CBS 117226, *A. ganisen* CBS 118488, and the *A. infectoria* isolate STE-U4271 (AF397248) as outgroups.

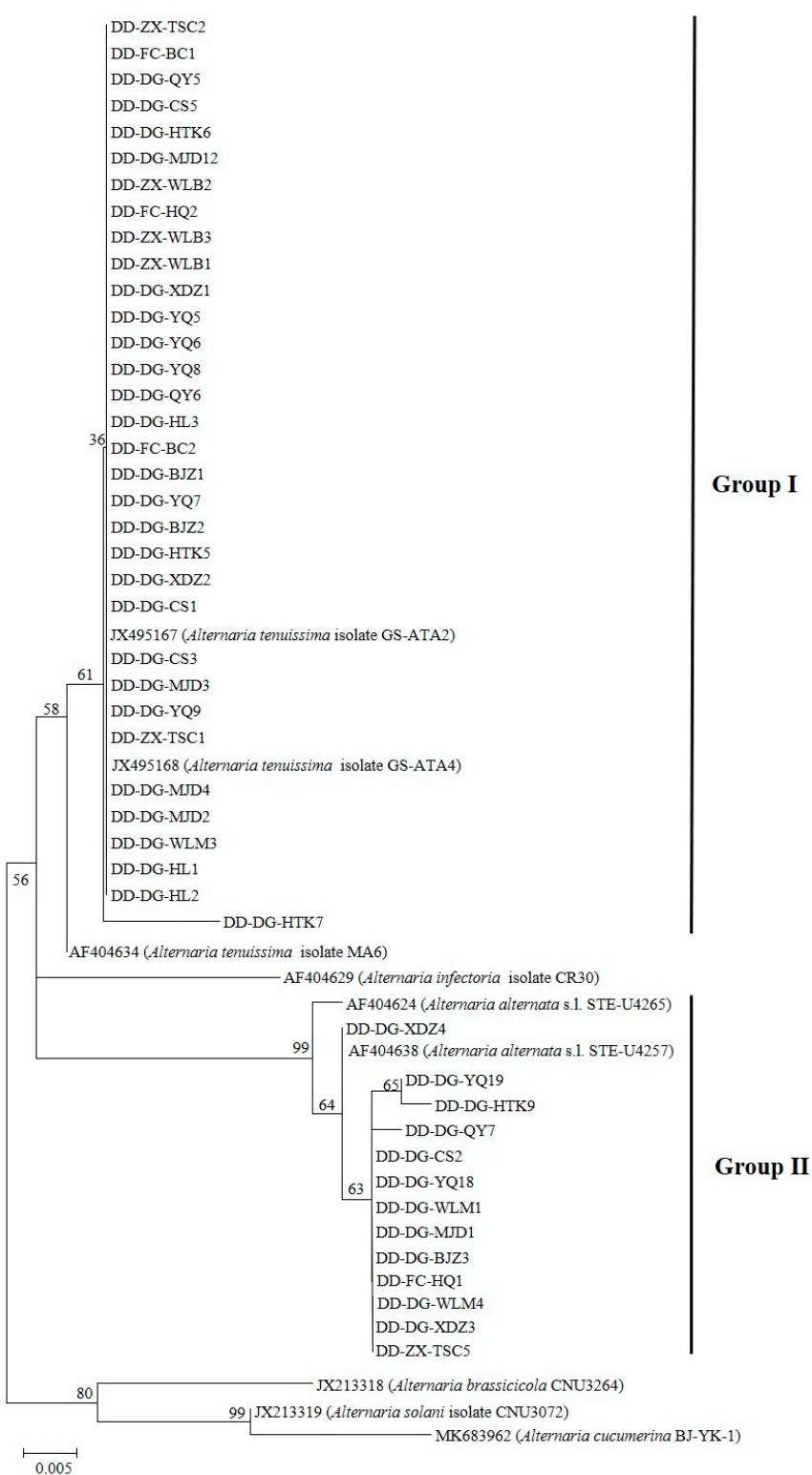

**Figure 3.** Phylogenetic tree based on the partial coding sequences of the histone 3 gene of the 46 *Alternaria* isolates used in pathogenicity and the 9 reference sequences published in the NCBI library. The algorithm is based on the Maximum Likelihood estimate. Bootstrap values with 1000 replications were found at nodes on the phylogenetic tree, which identified the *A. brassicicola* CNU3264, *A. solani* isolate CNU3072, *A. cucumerina* BJ-YK-1, and the *A. infectoria* isolate CR30 (AF404634) as outgroups.

### 3.2. Pathogenicity Analysis

After two weeks, the detached strawberry leaves that had been inoculated with the *Alternaria* isolates were crimped, which produced marked wheeling, dark brown lesions, and even the typical yellow halo (Figure 4). Samples of the diseased tissues were re-isolated to assess them for Koch's postulates. The incidence rates of disease on detached strawberry leaves caused by *A. tenuissima* were 49.6–100.0%, and the disease indexes were 20.4–48.7 (Table 2). The disease incidence and DI values of leaves inoculated with *A. alternata* were 67.2–100.0% and 32.2–59.5, respectively. Isolates of *A. tenuissima* and *A. alternata* generally exhibited a relatively high level of virulence on strawberry leaves.

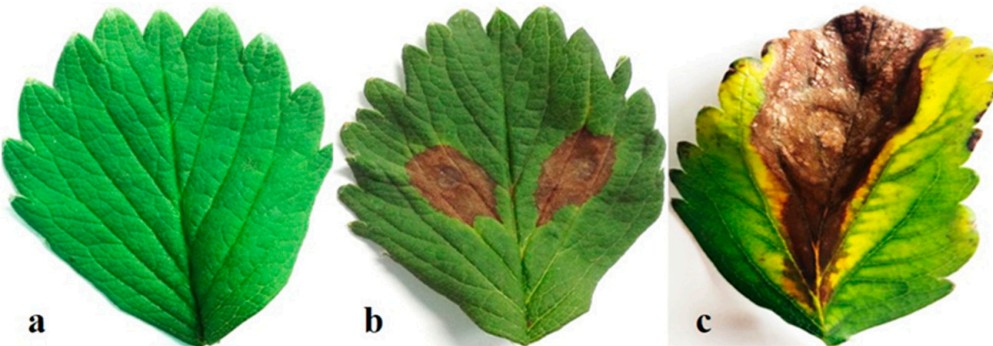

**Figure 4.** Pathogenicity of the 46 *Alternaria* isolates on healthy detached strawberry leaves of the variety "Dandong 99". (**a**): inoculated with sterile water (control); (**b**): inoculated with *A. tenuissima*; (**c**): inoculated with *A. alternata*.

**Table 2.** Disease incidence and disease index values of 46 *Alternaria* isolates when placed on detached strawberry leaves.

| *Alternaria* Species | Isolates Number | Disease Incidence (%) [Mean ± Standard Deviation (Range)] | Disease Index [Mean ± Standard Deviation (Range)] |
|---|---|---|---|
| *A. tenuissima* | 33 | 46.5 ± 3.7 (49.6–100.0) [b] | 30.6 ± 0.8 (20.4–48.7) [b] |
| *A. alternata* | 13 | 75.6 ± 6.2 (67.2–100.0) [a] | 47.2 ± 1.7 (32.2–59.5) [a] |

[a,b] Agar plugs were placed over a small puncture wound on the upper surface of a detached leaf obtained from the strawberry variety "Dandong 99". DS was scored after 14 days of incubation at 25 °C and 90% relative humidity. Disease incidence and disease index values followed by different letters are significantly different ($p < 0.05$). The data of disease incidence were analyzed by one-way analysis of variance (ANOVA) with Dunnett's T3 tests, and the data of the disease index were analyzed by one-way ANOVA with least significant difference (LSD) tests.

## 4. Discussion

In this study, two *Alternaria* species were identified as causal agents of strawberry black spot disease in the main strawberry-producing regions of China. These results were consistent with a report by Fu et al. (2020), who confirmed that *A. tenuissima* and *A. alternata* were the pathogens causing strawberry black spot disease in Beijing. *A. tenuissima* was the dominant fungus in both Beijing and Dandong, China. The complex isolates of *A. alternata* and *A. tenuissima* were single isolates obtained from different plants, such as potato [21], watermelon [22], muskmelon [23], and Chinese cabbage [24]. Companion planting is a traditional practice that aims to attract pollinators and, thus, increase the yield and quality of strawberries by planting a co-flowering species nearby [25]. However, the strawberry companion plants listed do not include potato, watermelon, muskmelon, and Chinese cabbage, which could interact negatively with strawberry.

The *Alternaria* species recognized as the pathogenic fungus has more than 4000 recognized members, including nearly 300 species found globally [4]. Based on phylogenetic and morphological studies, the *Alternaria* species are currently divided into 26 categories [26]. In this study, the morphological characteristics of 108 isolates obtained from strawberry

leaves with black spot disease were consistent with the characteristics of *A. tenuissima* and *A. alternata* as described by Simmons [17]. The species identified from the 108 isolates was *A. consortialis*, based on ITS rDNA sequence analysis. Based on the combined datasets of ITS rDNA, SSU, *gpd*, *rpb2*, *tef1*, histone 3 gene, ATPase, and *endoPG*, *A. tenuissima* and *A. alternata* were found to have similar loci as were found in previous observations [4,26,27]. Glass and Donaldson [28] stated that the histone 3 gene could be used in the phylogenetic analysis and detection of filamentous ascomycetes. Kang et al. [29] used the histone 3 gene combined with the ITS 1 and 2 regions of the rRNA gene to separate the *Alternaria* species into five clades: *Alternaria sp.*, *A. arborescens*, *A. infectoria*, *A. tenuissima*, and one clade containing isolates with a variable morphology. Shi et al. [24] used the histone 3 gene and ITS rDNA to distinguish four subsets of the *Alternaria* species causing leaf spot on Chinese cabbages in Shanxi province of China: *A. tenuissima*, *A. alternata*, *A. brassicicola*, and *A. solani*. The previous results show that, based on the partial coding sequences of the histone 3 gene, the *A. tenuissima* and *A. alternata* isolates were clustered into two distinct clades. The histone 3 gene was identified as a useful tool for distinguishing *Alternaria* species.

*A. tenuissima* and *A. alternata* can produce a wide spectrum of secondary metabolites, such as alternariol, AOH, AME, TeA, TEN, and ALT [30]. *Alternaria* toxins have the potential to threaten strawberry production, especially given their harmful effects on human health [31]. The pathogenicity tests prove that the incidence of strawberry black spot disease caused by *Alternaria* isolates was up to 100% in the main strawberry-producing area of Dandong. Further studies on *Alternaria* isolates taken from strawberry plants with black spot disease should be undertaken to develop green, safe, and effective fungicides, thus eliminating the potential harm *Alternaria* toxins can cause in the human body.

## 5. Conclusions

In this study, we reported 108 pathogenic fungi causing strawberry black spot disease in Dandong. These isolates were identified as *A. tenuissima* (78 isolates, 72.2%) and *A. alternata* (30 isolates, 27.8%). To date, this paper is the first to describe *A. tenuissima* and *A. alternata* as the causative agents of black spot disease in strawberry plants in Dandong, China.

**Supplementary Materials:** The following supporting information can be downloaded at: https://www.mdpi.com/article/10.3390/agronomy13041014/s1, Table S1: *Alternaria* isolates used for phylogenetic analysis.

**Author Contributions:** Conceptualization, X.S. and C.W.; methodology, X.G.; resources, writing—original draft preparation, X.S. and C.W.; writing—review and editing, Y.F.; supervision, X.W. All authors have read and agreed to the published version of the manuscript.

**Funding:** This research was funded by the science and technology research projects of the education department of Liaoning province (LNASNM201903).

**Institutional Review Board Statement:** Not applicable.

**Informed Consent Statement:** Not applicable.

**Data Availability Statement:** The data presented in this study are available on request from the corresponding author.

**Conflicts of Interest:** Not applicable.

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
