# Peer review of "Characterization of Alternaria Species Associated with Black Spot of Strawberry in Dandong, China"

_agronomy, doi:10.3390/agronomy13041014_

Round 1

Reviewer 1 Report (New Reviewer)

  • Line 139-141: What the authors mean by "After amplification by the histone 3 gene primer, the PCR products of 108 isolates were divided 140 into two fragments: 546 bp (78 isolates) and 440 bp (30 isolates)".
  • Supplementary Table 1: must include latitude and longitude location, not general location
  • language needs to be improved.
  • Line 145: The authors mentioned that "phylogeny trees were created for 47 Alternaria isolates used in pathogenicity experiments based on ITS sequence and the histone 3 gene data. The figures 2 and 3 ligands as well as materials and methods (line 92) mention 46 isolates, although the phylogenetic tree presents 48 isolates. Please clarify and correct. There were 46, 47, or 48 isolates.
  • The ITS gene sequence alone, as well as the histone 3 gene sequence, is not informative. Authors need to sequence an additional reference gene such as GAPDH, rpb2, or tef1. Otherwise, the new concatenated phylogenetic tree based on both sequences must be presented rather than a separate phylogeny for each.Furthermore, the phylogenetic tree must include all Alternaria species, not just two or three 
  • Line 178: Under Table 2, the author wrote an asterisk (*). What does that sign refer to?
  • Table 2: I see that the lowest average numbers (63.06.4 and 45.66.9) for disease incidence (%) and disease index (81.28.8 and 47.21.7) had letter "a," while the highest numbers (81.28.8 and 47.21.7) had letter "b."Is this right? Please check again.
  • Line 187: "Those results were consistent with Fu et al. (2020) report." Change to "These results were consistent with Fu et al. (2020) report."

Author Response

Thank you for your valuable and thoughtful comments. We have carefully checked and improved the English writing in the revised manuscript

  1. Line 139-141: After amplification by the histone 3 gene primer, the PCR products of 108 isolates were divided into two fragments: 546 bp (78 isolates) and 440 bp (30 isolates).

Electrophoretic examination revealed that the PCR amplification of the partial coding sequences of the histone 3 gene was divided into two parts. Compared with standard marker, their lengths are 546 bp and 440 bp, respectively.

Supplementary Table 1: We have added latitude and longitude location.

  1. Line 145:

Thank you for pointing out this problem in our manuscript. According to the revised content, we have unified 46 isolates in revised paper. Also, we have added other Alternaria species and redrawn Figure 2 and 3.

We agree that the recent revision of the Alternaria section Alternaria by Woudenberg et al (2015) based on the analysis of genome and transcriptome comparisons and molecular phylogenies, which synonymized the A. tenuissima under Alternaria sect. Alternaria alternata group. Using the combined datasets of ITS rDNA, SSU, gpd, rpb2, tef1, histen 3 gene, ATPase and endoPG, A. tenuissima and A. alternata had similar loci to previous observations (Woudenberg et al., Stud. Mycol. 2015, 82, 1-21; Lawrence et al., Mycol. Prog. 2016, 15, 1-22; Bessadat et al., Life (Basel) 2021, 11, 1291.). In this group, the histone 3 gene was useful in distinguishing some closed species of Alternaria especially the small-spored species, A. tenuissima and A. alternata. In the past ten years, our lab has conducted a series of studies on Alternaria species that are the causal agents of foliar diseases on a variety of crops, including potato, sunflower, watermelon, and muskmelon. In these publications, we have confirmed that the fragment length of the histone 3 gene is useful for distinguishing between some species of Alternaria, especially the small-spored species, A. tenuissima and A. alternata. Here are a few of the representative publications from our lab on this topic.

(1) Wang TY, Zhao J, Sun P, Wu XH. 2014. Characterization of Alternaria species associated with leaf blight of sunflower in China. European Journal of Plant Pathology, 140, 301-315.

(2) Zheng HH, Zhao J, Wang TY, Wu XH. 2015. Characterization of Alternaria species associated with potato foliar diseases in China. Plant Pathology, 64, 425-433.

(3) Zhao J, Bao SW, Ma GP, Wu XH. 2016. Characterization of Alternaria species associated with watermelon leaf blight in Beijing municipality of China. Journal of Plant Pathology, 98, 135-138.

(4) Zhao J, Bao SW, Ma GP, Wu XH. 2016. Characterization of Alternaria species associated with muskmelon foliar diseases in Beijing municipality of China. Journal of General Plant Pathology, 82, 29-32.

(5) Zhao J, Ma GP, Liu YY, Wu XH, 2018. Alternaria species infecting potato in southern China. Canadian Journal of Plant Pathology, 40, 312-317.

(6) Wang TY, Zhao J, Ma GP, Bao SW, Wu XH, 2019. Leaf blight of sunflower caused by Alternaria tenuissima and A. alternata in Beijing, China. Canadian Journal of Plant Pathology,  https://doi.org/10.1080/07060661.2019.1583288.

(7) Fu Y, Zhang XF, Liu SJH, Hu KL, Wu XH, 2020. Characterization of Alternaria species associated with black spot of strawberry in Beijing municipality of China. Canadian Journal of Plant Pathology, 4, 235-242.

(8) Shi XJ, Zeng KX, Wang XY, Liang ZH, Wu XH, 2021. Characterization of Alternaria species causing leaf spot on Chinese cabbage in Shanxi province of China. Journal of Plant Pathology, 103, 289-293.

  In addition, there were some published papers which indicated that the histone 3 gene was useful in distinguishing some closed species of Alternaria.

(1) Glass NL, Donaldson GC. 1995. Develoliujiapment of primer sets designed for use with the PCR to amplify conserved genes from filamentous ascomycetes. Applied and Environmental Microbiology, 61, 1323-1330.

(2) Kang JC, Crous PW, Mchau GRA, Serdani M, Song SM. 2002. Phylogenetic analysis of Alternaria spp. associated with apple core rot and citrus black rot in South Africa. Mycological Research, 106, 1151-1162.

(3) Deng JX, Paul NC, Park MS, Yu SH. 2013. Molecular characterization, morphology, and pathogenicity of Alternaria panax from araliaceous plants in Korea. Mycological Progress, 12, 383-396.

(4) He MH, Li DL, Zhu W, Wu EJ, Yang LN, Wang YP, Waheed A, Zhan JS, 2017. Slow and temperature‐mediated pathogen adaptation to a nonspecific fungicide in agricultural ecosystem. Evolutionary Applications, 11, 182-192.

(5)  Landschoot S, Vandecasteele M, Baets BD, Höfte M, Audenaert K, Haesaert G, 2017. Identification of A. arborescens, A. grandis, and A. protenta as new members of the European Alternaria population on potato. Fungal Biology, 121, 172-188.

(6) Vandecasteele M, Landschoot S, Carrette J, Verwaeren J, Höfte M, Audenaert K, Haesaert G, 2018. Species prevalence and disease progression studies demonstrate a seasonal shift in the Alternaria population composition on potato. Plant Pathology, 67, 327-336.

  1. Table 2: We have re-checked the calculation and modified Table 2.
  2. Line 187: Thank you for pointing out this problem in our manuscript. The sentence has been modified according to your suggestion.

Reviewer 2 Report (New Reviewer)

Title of study is quite relevant. Manuscript is well drafted. However, there is scope of improvements as there are many grammatical mistakes, syntax error and error in construction of sentences are observed. You are advised to get the manuscript checked by native English speaker of take help from Language Editing services

Author Response

Thank you for your suggestions. According to your advice, this manuscript was edited for proper English language, grammar, punctuation, spelling, and overall style by one or more of the highly qualified native English speaking editors at MDPI.

Reviewer 3 Report (New Reviewer)

Very preliminary experiments are doing here Author can do several other parameter related to disease content in host plants. 
Abstract is not catchy 
Alternaria is well studied pathogen author can get some molecular pathogenicity  
Introduction is winding and unfocussed
 Language issues

Author Response

Thank you for your suggestions. According to your advice, this manuscript was edited for proper English language, grammar, punctuation, spelling, and overall style by one or more of the highly qualified native English speaking editors at MDPI.

Reviewer 4 Report (Previous Reviewer 2)

Dear author,

I have previously stated that the ITS and histone regions are not sufficient for the differentiation of Alternaria species, adding different gene (GAPDH, RPB2, TEF, Alt a 1) sequences. Despite this, the author insists that morphological characteristics make the distinction between species and support this with ITS and histone gene regions. Frankly, I am concerned that closely related species in the genus Alternaria distinguish between these two genes. Kind regards

Author Response

Thank you for your valuable and thoughtful comments. This manuscript was edited for proper English language, grammar, punctuation, spelling, and overall style by one or more of the highly qualified native English speaking editors at MDPI.             

We agree that the recent revision of the Alternaria section Alternaria by Woudenberg et al (2015) based on the analysis of genome and transcriptome comparisons and molecular phylogenies, which synonymized the A. tenuissima under Alternaria sect. Alternaria alternata group. Using the combined datasets of ITS rDNA, SSU, gpd, rpb2, tef1, histen 3 gene, ATPase and endoPG, A. tenuissima and A. alternata had similar loci to previous observations (Woudenberg et al., Stud. Mycol. 2015, 82, 1-21; Lawrence et al., Mycol. Prog. 2016, 15, 1-22; Bessadat et al., Life (Basel) 2021, 11, 1291.). In this group, the histone 3 gene was useful in distinguishing some closed species of Alternaria especially the small-spored species, A. tenuissima and A. alternata. In the past ten years, our lab has conducted a series of studies on Alternaria species that are the causal agents of foliar diseases on a variety of crops, including potato, sunflower, watermelon, and muskmelon. In these publications, we have confirmed that the fragment length of the histone 3 gene is useful for distinguishing between some species of Alternaria, especially the small-spored species, A. tenuissima and A. alternata. Here are a few of the representative publications from our lab on this topic.

(1) Wang TY, Zhao J, Sun P, Wu XH. 2014. Characterization of Alternaria species associated with leaf blight of sunflower in China. European Journal of Plant Pathology, 140, 301-315.

(2) Zheng HH, Zhao J, Wang TY, Wu XH. 2015. Characterization of Alternaria species associated with potato foliar diseases in China. Plant Pathology, 64, 425-433.

(3) Zhao J, Bao SW, Ma GP, Wu XH. 2016. Characterization of Alternaria species associated with watermelon leaf blight in Beijing municipality of China. Journal of Plant Pathology, 98, 135-138.

(4) Zhao J, Bao SW, Ma GP, Wu XH. 2016. Characterization of Alternaria species associated with muskmelon foliar diseases in Beijing municipality of China. Journal of General Plant Pathology, 82, 29-32.

(5) Zhao J, Ma GP, Liu YY, Wu XH, 2018. Alternaria species infecting potato in southern China. Canadian Journal of Plant Pathology, 40, 312-317.

(6) Wang TY, Zhao J, Ma GP, Bao SW, Wu XH, 2019. Leaf blight of sunflower caused by Alternaria tenuissima and A. alternata in Beijing, China. Canadian Journal of Plant Pathology,  https://doi.org/10.1080/07060661.2019.1583288.

(7) Fu Y, Zhang XF, Liu SJH, Hu KL, Wu XH, 2020. Characterization of Alternaria species associated with black spot of strawberry in Beijing municipality of China. Canadian Journal of Plant Pathology, 4, 235-242.

(8) Shi XJ, Zeng KX, Wang XY, Liang ZH, Wu XH, 2021. Characterization of Alternaria species causing leaf spot on Chinese cabbage in Shanxi province of China. Journal of Plant Pathology, 103, 289-293.

  In addition, there were some published papers which indicated that the histone 3 gene was useful in distinguishing some closed species of Alternaria.

(1) Glass NL, Donaldson GC. 1995. Develoliujiapment of primer sets designed for use with the PCR to amplify conserved genes from filamentous ascomycetes. Applied and Environmental Microbiology, 61, 1323-1330.

(2) Kang JC, Crous PW, Mchau GRA, Serdani M, Song SM. 2002. Phylogenetic analysis of Alternaria spp. associated with apple core rot and citrus black rot in South Africa. Mycological Research, 106, 1151-1162.

(3) Deng JX, Paul NC, Park MS, Yu SH. 2013. Molecular characterization, morphology, and pathogenicity of Alternaria panax from araliaceous plants in Korea. Mycological Progress, 12, 383-396.

(4) He MH, Li DL, Zhu W, Wu EJ, Yang LN, Wang YP, Waheed A, Zhan JS, 2017. Slow and temperature‐mediated pathogen adaptation to a nonspecific fungicide in agricultural ecosystem. Evolutionary Applications, 11, 182-192.

(5)  Landschoot S, Vandecasteele M, Baets BD, Höfte M, Audenaert K, Haesaert G, 2017. Identification of A. arborescens, A. grandis, and A. protenta as new members of the European Alternaria population on potato. Fungal Biology, 121, 172-188.

(6) Vandecasteele M, Landschoot S, Carrette J, Verwaeren J, Höfte M, Audenaert K, Haesaert G, 2018. Species prevalence and disease progression studies demonstrate a seasonal shift in the Alternaria population composition on potato. Plant Pathology, 67, 327-336.

Round 2

Reviewer 1 Report (New Reviewer)

The authors have addressed all of my concerns with the original manuscript. The revised manuscript is ready for publication.

Reviewer 3 Report (New Reviewer)

Best of luck for upcoming paper and future science 

Reviewer 4 Report (Previous Reviewer 2)

Dear Author

I have previously stated that the ITS and histone regions are not sufficient for the differentiation of Alternaria species, adding different gene (GAPDH, RPB2, TEF, Alt a 1) sequences. Despite this, the author insists that morphological features make the distinction between species and support this with ITS and histone gene regions. Frankly, I'm concerned that closely related species in the genus Alternaria distinguish between these two genes. Kind regards

This manuscript is a resubmission of an earlier submission. The following is a list of the peer review reports and author responses from that submission.

Round 1

Reviewer 1 Report

The paper is interesting but it needs more revision before to submit to the Journal 

The abstract should have some value  of your results

Keyword need to modification and chose different than in title 

Introduction need more references see attach file 

Many scientific name need to be italic 

You wrote Disease severity (DS) and Disease index (DI) in material but in results you mentioned Disease incidence 

Nearly I did not see any discussion in the paper 

References need to update and format according to the Journal 

Reviewer 2 Report

Dear author, 

In the study, 108 Alternaria isolates obtained from the Dandong that the largest strawberry growing province of China were identified with morphological and molecular characteristics. The authors amplified ribosomal DNA ITS regions and the partial coding sequence of the histone 3 gene in molecular separation of isolates. However, these regions used is insufficient to distinct Alternaria species especially between A. alternata and A. tenuissima. Sequence analysis of glyceraldehyde- 3-phosphate dehydrogenase (GAPDH), RNA polymerase second largest subunit (RPB2), translation elongation factor (TEF), and Alternaria major allergen (Alt a 1) genes of the isolates should be performed. 

Best regards

Reviewer 3 Report

I do not sure this manuscript suits to the scope of the Agronomy journal. See subject areas below:

  • Crop breeding and genetics
  • Chemistry, biology, and genetics applied to agronomy
  • Biotechnology for farming and the use of plants, plant breeding
  • Farming and cropping systems
  • Precision agriculture
  • Crop-livestock interactions
  • Crop and soil interactions
  • Soil heath and plant nutrition for sustainable agriculture
  • Agronomy of urban and peri-uban areas
  • Organic farming
  • Weed science and weed management systems
  • Industrial and bioenergy crops
  • Horticultural and floricultural crops
  • Agroecosystems and the environment
  • Sustainable development of agronomy
  • Sustainability, biodiversity and ecosystem services of bioenergy cropping systems
  • Crop physiology
  • Water management/Irrigation in agronomy
  • Agricultural meteorology (climate change)
  • Grassland and pasture improvement and agronomy
  • Food systems

At the same time, the manuscript describes the study of strawberry pathogens from the genus Alternaria. It does not contain any parts related to the agronomy or at least crop protection techniques... just characterization and species identification of isolated strains with pathogenicity tests on detached leaves. No any fungicide resistance studies, or protection schemes. In my opinion, this manuscript should be submitted rather to the more suitable journal, such as, for example, Pathogens, Microorganisms, etc., but not to the Agronomy journal.

In addition, I would strongly recommend authors to perform language editing, since some parts of the text looks unclear or contain some language errors.

Other comments are listed below.

 Title

"First report of Alternaria sp. causing black spot on strawberry in Dandong, China" - usually such type of titles (first report of ... any pathogen in any location") mean that this pathogen was first observed in this location for any time period. I do not sure it is suitable for your case since the situation seems to be slightly different. You did not describe a pathogen, which was not observed in this location for years, but then was suddenly revealed (due to climatic changes, or spreading from neighboring regions, or import with some seeds or plants, etc.), but rather write that no studies of such kind were performed in this location before (i.e., this pathogen could occur there for years prior you arranged your study). I would recommend to change the title of this manuscript (see, for example, the paper of Fu et al. cited in this manuscript - they used a neutral title, but mentioned that was the first study of such kind at the end of the Intro section).

Introduction

"...more than 200,000 mu" - please, give the area in standard units.

If possible, it would be better to give references for the production volumes and other numbers characterized strawberry production in the province.

Materials and methods

Subsection 2.2: Please, add information about the process of DNA extraction as I see no text about this.

"the partial coding sequence of the histone 3 gene was confirmed the Alternaria species" - why did you choose this gene? As I understand, this sequence was used to distinguish A. alternata from A. tenuissima. Please, give a reference.

Subsection 2.3: please, give more detailed description of the assay. You write about agar disks used for inoculation of detached leaves. However, you did not mention, for example, how long pathogen cultures were grown on agar prior inoculation. Were the disks taken from the colony edges or from central parts? How the inoculation process occurred: at which part of a leaf did you out agar discs and how long they remained there (for all 14 days or not)?

Though you give a reference to Fu et al. in relation to the calculation of disease indices and disease severity, it would be good to briefly describe what these parameters mean and give the formulas.

Results

Fig. 1: please indicate, which pictures describe A. alternata and which - A. tenuissima. Now the figure caption does not contain this information. As I understand, one species is shown in the left part of the figure, and second is shown in the right part.

"Selected A. tenuissima (KR867207) and A. alternata (GU566303, KR867035, and MH862229) as variables, it has over 99% homology to." - this sentence should be re-phrased as its sense is rather unclear.

Table 2: in the Note you write about lowercase letter to indicate the significant difference between values. Actually, no such letters are in the table. Moreover, if data in parentheses indicate mean ± SD, then how did you get means equal to 65.6 and 67.2 for disease indices of both Alternaria species, if the DI range was 20.4-48.7 and 32.2 - 59.5, i.e., was below the mean value? Please, check and correct.